# Divergent evolutionary trajectories shape the postmating transcriptional profiles of conspecifically and heterospecifically mated cactophilic *Drosophila* females

Fernando Diaz [1,5✉], Carson W. Allan [1], Xingsen Chen[1], Joshua M. Coleman[1], Jeremy M. Bono [2✉] & Luciano M. Matzkin [1,3,4✉]

Postmating-prezygotic (PMPZ) reproductive isolation is hypothesized to result from divergent coevolutionary trajectories of sexual selection and/or sexual conflict in isolated populations. However, the genetic basis of PMPZ incompatibilities between species is poorly understood. Here, we use a comparative framework to compare global gene expression in con- and heterospecifically mated *Drosophila mojavensis* and *D. arizonae* female reproductive tracts. We find striking divergence between the species in the female postmating transcriptional response to conspecific mating, including differences in differential expression (DE), alternative splicing (AS), and intron retention (IR). As predicted, heterospecific matings produce disrupted transcriptional profiles, but the overall patterns of misregulation are different between the reciprocal crosses. Moreover, we find a positive correlation between postmating transcriptional divergence between species and levels of transcriptional disruption in heterospecific crosses. This result indicates that mating responsive genes that have diverged more in expression also have more disrupted transcriptional profiles in heterospecifically mated females. Overall, our results provide insights into the evolution of PMPZ isolation and lay the foundation for future studies aimed at identifying specific genes involved in PMPZ incompatibilities and the evolutionary forces that have contributed to their divergence in closely related species.

[1] Department of Entomology, University of Arizona, Tucson, AZ, USA. [2] Department of Biology, University of Colorado Colorado Springs, Colorado Springs, USA. [3] BIO5 Institute, University of Arizona, Tucson, AZ, USA. [4] Department of Ecology and Evolutionary Biology, University of Arizona, Tucson, AZ, USA. [5] Present address: Biology Department, Colgate University, Hamilton, NY, USA. ✉email: ferdiazfer@gmail.com; jbono@uccs.edu; lmatzkin@arizona.edu

Speciation results from the accumulation of reproductive isolating barriers that evolve as a consequence of genetic divergence between isolated populations[1]. While historically most speciation research has focused on barriers that arise before mating (premating) or after fertilization (postzygotic), there has been increasing recognition that postmating-prezygotic (PMPZ) isolation is a potent and rapidly evolving barrier to gene flow in many taxa[2,3]. PMPZ isolation arises as a result of incompatible interactions between male and female gametes or among reproductive molecules involved in postcopulatory interactions. The rapid evolution of PMPZ isolation is consistent with the well-established pattern that genes involved in postcopulatory interactions are among the most rapidly evolving in the genome of many internally fertilizing organisms[4].

Models of the evolution of PMPZ isolation in animals typically assume that rapid divergence of genes involved in postcopulatory interactions arises due to divergent trajectories of sexual selection and sexual conflict in isolated populations[2,3]. This is particularly likely in species where females remate frequently, because selection on traits mediating male fertilization success and female choice is expected to be intense[5]. Male-female coevolutionary dynamics may follow different paths in diverging populations, leading to alterations in postcopulatory molecular processes that ultimately give rise to PMPZ incompatibilities in crosses between populations[2,3]. Despite growing evidence highlighting the importance of PMPZ isolation in driving the speciation process, the molecular underpinnings of PMPZ incompatibilities in internally fertilizing animals are not well understood[2].

Although rapid divergence in postmating molecular interactions between the sexes may lead to the evolution of PMPZ isolation, data comparing postmating responses between closely related species pairs is limited. Nevertheless, a recent comparative proteomic analysis of *Drosophila simulans* and *D. mauritiana* revealed extensive species divergence in the proteome of both virgin and mated female reproductive tracts[6]. Only a few studies in plants and animals have explored the molecular basis of PMPZ incompatibilities by comparing postmating transcriptome or proteome responses of con- and heterospecifically-mated female reproductive tissues[7–9]. Heterospecific mating resulted in disruption of the female postmating transcriptional response in *D. mojavensis and D. novamexicana*[7,8]. These disruptions included subsets of genes associated with the normal transcriptional response to conspecific mating, and many additional genes that were not differentially regulated in response to mating in conspecifics[7,8]. In contrast, McCullough et al.[6] found few differences in the proteomic response to mating with conspecifics or heterospecifics in *D. simulans* females despite substantial evidence for PMPZ isolation in this cross. Interestingly, female genes that were transcriptionally misregulated in response to heterospecific mating generally do not evolve more rapidly at the sequence level than other genes in the genome and evolve at lower rates than male seminal fluid protein genes[8,10]. Overall, the relative paucity of comparative data on the molecular postmating response in species pairs isolated by PMPZ barriers highlights the need for additional studies. Study systems in which reciprocal heterospecific crosses can be investigated would be particularly valuable since this approach allows the mechanisms of PMPZ isolation to be compared. Sexual selection and sexual conflict are not necessarily expected to target the same set of traits in isolated populations[11]. While this leads to the prediction that the underlying molecular basis of reproductive incompatibilities in reciprocal crosses may not be the same, this has not been rigorously tested due to a lack of studies specifically comparing reciprocal crosses.

In this study, we use the *D. mojavensis/D. arizonae* study system to examine female postmating transcriptional responses in a comparative context. *Drosophila mojavensis* and *D. arizonae* are recently diverged sister species that have long been the focus of speciation research[12]. Previous studies have documented strong PMPZ isolation in crosses involving *D. mojavensis* females, which results in extremely low fertilization success following heterospecific copulation[13]. While the mechanistic basis of these incompatibilities is not fully understood, heterospecifically-mated females fail to efficiently degrade the insemination reaction and exhibit sperm storage defects[13]. The insemination reaction is a large, opaque mass that fills the uterus and typically dissolves over the course of several hours following conspecific matings[13–17]. In contrast to other reproductive isolating barriers that vary in magnitude depending on the source population of *D. mojavensis* females[12], the high strength of PMPZ isolation is consistent for *D. mojavensis* females from each of four geographically distinct populations[13], suggesting it may have evolved early in the process of divergence. Given that there is only limited descriptive data available on PMPZ isolation in crosses between *D. arizonae* females and *D. mojavensis* males[12,18], here we demonstrate the presence of strong PMPZ isolation in this cross. We then use a comparative framework to analyze postmating transcriptomic responses in con- and heterospecifically-mated females for both species. In addition to an analysis of differential expression (DE), we also examine the potential role of alternative splicing (AS) and intron retention (IR) in the postmating transcriptomic response in con- and heterospecifically-mated females. Although comparisons across *Drosophila* show that AS is particularly prevalent in reproductive tissues and contributes significantly to lineage-specific evolution[19], AS has not been considered in previous studies of the postmating response in female reproductive tissues. We recently demonstrated considerable postmating AS in heads of con- and heterospecifically-mated females[20], suggesting that AS may play an underappreciated role in the female postmating response. Moreover, disruption of typical patterns of AS may represent an additional mechanism resulting in PMPZ incompatibilities.

## Results

**Postmating-prezygotic isolation is strong in crosses between *D. arizonae* females and *D. mojavensis* males.** PMPZ isolation between *D. mojavensis* and *D. arizonae* was previously documented in crosses involving *D. mojavensis* females from four geographically distinct populations[13]. Fecundity was lower in three of the four crosses and fertilization success was dramatically reduced in all crosses[13]. These incompatibilities appear to be related to problems with sperm storage and failure to efficiently degrade the insemination reaction in heterospecifically-mated females[13]. Here we test for PMPZ isolation in the reciprocal cross by examining fecundity, fertilization success, and sperm storage. Heterospecifically-mated *D. arizonae* females laid 23% fewer eggs over the course of seven days compared to conspecifically mated females (t-test, $t = 2.15$, $P = 0.04$; Fig. 1a), and a much smaller proportion of eggs laid by heterospecifically mated females hatched (GLMM, $X^2 = 26.9$, $P < 0.001$; Fig. 1b). Analysis of DAPI stained eggs indicated that most unhatched eggs laid by heterospecifically mated females were likely unfertilized (GLM, $X^2 = 242.9$, $P < 0.001$; Fig. 1c). We acknowledge the caveat that differentiating unfertilized eggs from inviable embryos arrested during the earliest syncytial divisions may not be possible. However, of the approximately 30% of embryos that were clearly developing, all were at stages of embryogenesis consistent with their age (i.e. >6 h). We did not observe embryos arrested at a range of timepoints between the earliest syncytial divisions and six hours as might be expected if inviability was common. Given this, we conclude that the most likely explanation for low larval

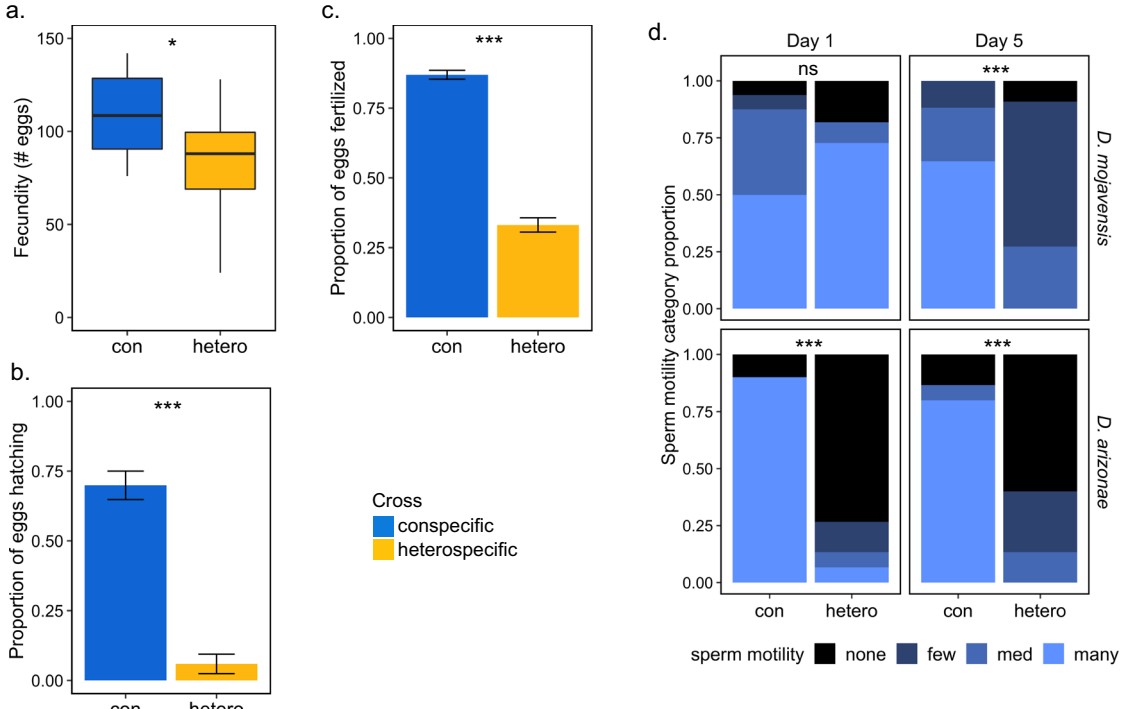

**Fig. 1 Evidence of postmating-prezygotic isolation between *D. mojavensis* and *D. arizonae*. a** Egg production over seven days of *D. arizonae* females following a con- or heterospecific mating (*n* = 10 and 11 females, respectively). **b** Proportion of eggs oviposited by *D. arizonae* females in **a**, that hatch into first instar larva. **c** Proportion of eggs oviposited by *D. arizonae* females following a con- or heterospecific mating that were determined to be fertilized using DAPI staining (*n* = 445 and 338 eggs, respectively). These results indicate an overall decrease in reproductive success of heterospecifically mated *D. arizonae*. **d** Levels of sperm motility within seminal receptacles in both *D. mojavensis* and *D. arizonae* females mated con- and heterospecifically at 1 and 5 days postmating. *P*-values of correlations are noted: *$P < 0.05$, **$P < 0.01$, ***$P < 0.001$, ns not significant ($P > 0.05$). Boxplots represent the median with 25th and 75th percentiles, and whiskers show the 1.5 interquartile range. Bar graphs identify mean and standard error bars.

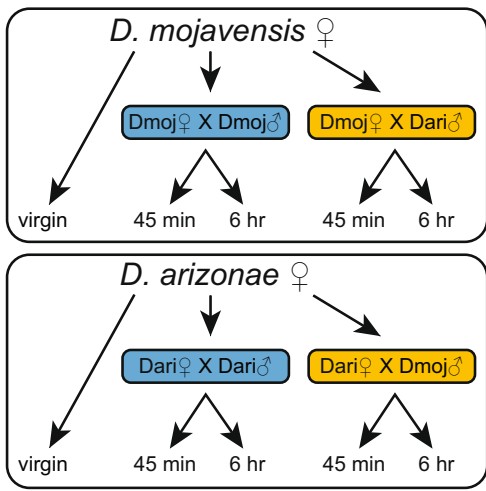

**Fig. 2 Experimental design used for differential expression and alternative splicing as a response to conspecific (blue) and heterospecific (yellow) matings in *D. mojavensis* and *D. arizonae*.** RNA-seq libraries were constructed from LRT of virgins, con- and heterospecifically mated females at 45 min and 6 h postmating. All comparisons were performed between virgin females and mated females for each species.

hatching is a reduction in fertilization success. The effect of heterospecific mating on sperm storage was dramatic in *D. arizonae* females where the combined "many" and "medium" motility categories fell from ~85% in conspecifically mated females to ~13% in heterospecifically mated females at both one

and five days post-mating ($X^2 = 17.8$, $P < 0.001$ and $X^2 = 20.1$, $P < 0.001$, respectively; Fig. 1d). A similar decline in sperm motility was observed in *D. mojavensis* at five days post-mating ($X^2 = 16.6$, $P < 0.001$), but no significance was observed on day one (Fig. 1d). All PMPZ data is provided in Supplementary Data 1.

**RNA-seq analysis**. The design of the transcriptomic analysis of conspecific and heterospecific matings for both *D. mojavensis* and *D. arizonae* is shown in Fig. 2. We obtained an average of $20 \times 10^6$ mapped reads for each library following trimming and filtering of sequence reads (Supplementary Table 1). On average, approximately 83% of total reads uniquely mapped for each library. This mapping rate was consistent across mating experiments and time points in both species' genomes (Supplementary Table 1). Minimum count filtering was applied independently to all gene subfeatures at the beginning of each analysis (*e.g.*, exon, junction, intron). Summary data showing the number of differentially regulated genes for all cross/time combinations are shown in Supplementary Table 2, and the values for the DE, AS and IR analyses are reported in Supplementary Data 2.

**Transcriptional regulation in response to conspecific mating involves multiple mechanisms**. Conspecific mating in both species resulted in substantial transcriptional changes within female LRTs (Fig. 3a). These changes were detected for all types of gene regulation analyzed including DE, AS, and IR (Fig. 3a). Few genes were found to be differentially regulated by multiple mechanisms, indicating that DE, AS, and IR represent distinct postmating responses targeting different genes (Fig. 3a). In accordance with predictions based on the known role of IR as a

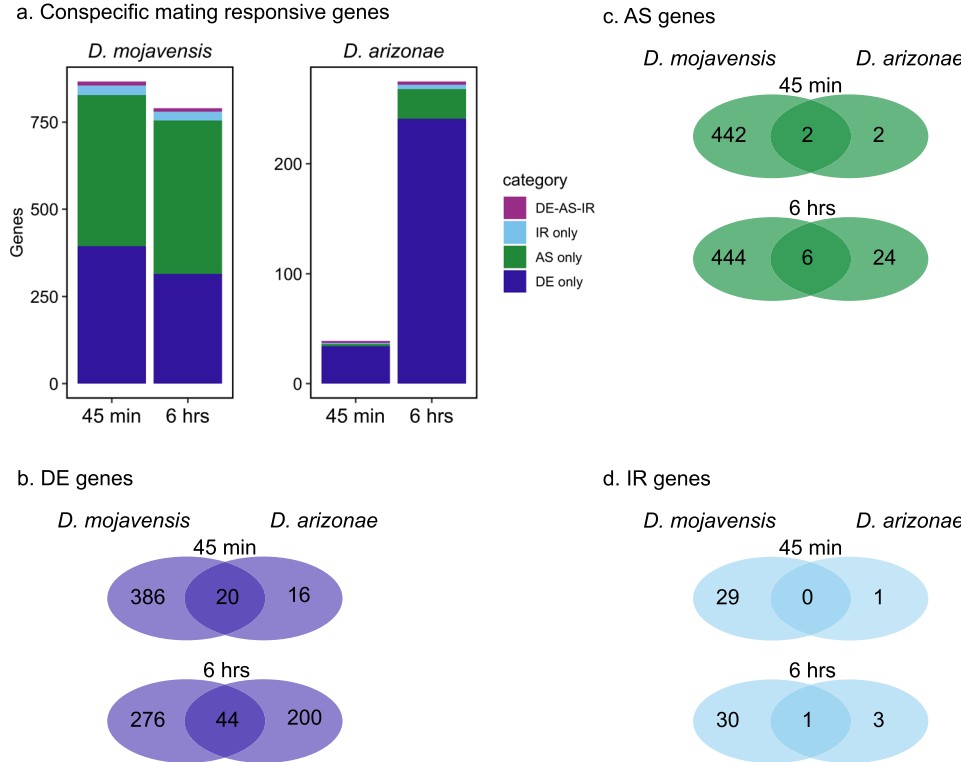

**Fig. 3 Conspecific postmating transcriptional response in *D. mojavensis* and *D. arizonae*.** **a** Number of conspecific mating responsive genes with significant patterns of DE, AS and IR following conspecific matings in *D. mojavensis* and *D. arizonae*. All comparisons were performed against virgin females at 45 min and 6 h postmating (FDR$_\alpha$ = 0.05). Genes in the DE-AS-IR category, showed significance in two or more of the individual categories (DE, AS and/or IR). Venn diagrams comparing significant **b** DE genes, **c** AS genes, and **d** IR genes from conspecific matings in each species. Overlapping genes represent the conserved postmating response, while species-specific genes indicate divergence in the postmating response. The distinct expression and splicing patterns indicate substantial differences in the postmating transcriptional response between the species.

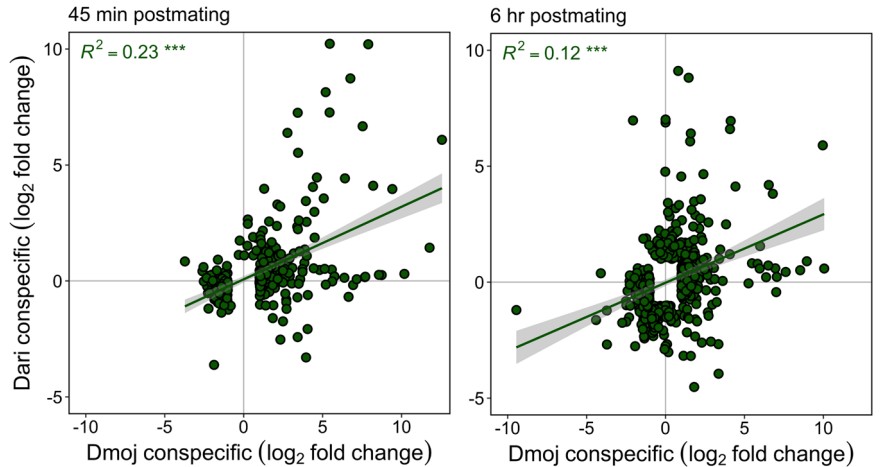

**Fig. 4 Transcriptional correlations between conspecific mating responsive DE genes in *D. mojavensis* vs *D. arizonae*.** Scatterplots represent the relationship between relative fold changes (log$_2$) for conspecific matings in *D. mojavensis* vs *D. arizonae* at 45 min and 6 h postmating. Log$_2$ fold changes are relative to virgin females. All conspecific mating responsive genes in each species are included (FDR$_\alpha$ = 0.05). Pearson's $R^2$ correlation coefficients and linear method trend-lines (with 95% confidence intervals shaded) between the species are indicated. *P*-values of correlations are noted: ****P* < 0.001.

mechanism of expression downregulation, we found that genes with higher intron retention were significantly downregulated compared to genes with lower rates of intron retention (Supplementary Fig. 1).

**Transcriptional response to mating has diverged between *D. mojavensis* and *D. arizonae*.** Quantitative transcriptional changes for conspecific mating responsive genes were significantly

positively correlated between the species ($R^2$ = 0.23, $P$ < 0.001 and $R^2$ = 0.12, $P$ < 0.001, at 45 min and 6 h respectively; Fig. 4). Moreover, the correlation for mating responsive genes was much higher than that for non-mating responsive genes (Supplementary Fig. 2). While these results indicate some conservation of the postmating transcriptional response between *D. mojavensis* and *D. arizonae*, most of the variation in the postmating response was not explained by the regression model (77% and 88% at 45 min

and 6 h respectively) and many genes displayed highly discordant patterns of expression between the two species (Fig. 4). Consistent with this, low overlap in the individual genes that were differentially regulated in response to mating in the two species indicates considerable divergence in the conspecific mating response (Fig. 3b–d). Moreover, the overall strength and temporal pattern of transcriptional response varied between the species. *Drosophila mojavensis* displayed a more rapid and stronger transcriptional response to conspecific mating that included substantial DE and AS. In contrast, in *D. arizonae* there was a much weaker initial transcriptional response to conspecific mating that increased over time. Furthermore, fewer genes were differentially expressed, and the amount of alternative splicing was significantly reduced compared to *D. mojavensis*.

GO-term enrichment analysis of DE genes revealed several overrepresented terms in conspecific crosses (Fig. 5). There were fewer enriched terms in *D. arizonae* than *D. mojavensis*, which likely reflects the overall smaller number of DE genes in this species. None of the enriched terms overlapped between the species, potentially indicating functional divergence in the postmating transcriptional response. GO-term enrichment analysis of AS genes did not detect any overrepresented categories in either conspecific cross.

**Transcriptional response to mating is disrupted in heterospecific crosses.** Heterospecific mating resulted in considerable changes in the transcriptome of the female LRT compared to virgin females, including differences involving DE, AS, and IR (Supplementary Fig. 3). Moreover, the normal conspecific postmating response in each species was significantly disrupted, as indicated by the relatively low number of genes that were differentially regulated in both con-and heterospecific crosses (Fig. 6). In *D. mojavensis*, although DE analysis revealed that most conspecific mating responsive genes exhibited disrupted expression in heterospecifically mated females at 45 min postmating (Fig. 6), the overall response between the crosses was still highly correlated ($R^2 = 0.83$, $P < 0.001$; Fig. 7a). Together, these results indicate that although the expression of most genes was significantly disrupted in the heterospecific cross, most of the differences in expression between the crosses were of small magnitude. At 6 h the proportion of genes with disrupted expression profiles remained high, and the magnitude of the differences also increased as indicated by a weaker correlation between crosses ($R^2 = 0.41$, $P < 0.001$; Fig. 7a). In *D. arizonae*,

most conspecific mating responsive genes also exhibited disrupted expression profiles (Fig. 6). However, in this case the magnitude of expression differences was greater at 45 min than 6 h ($R^2 = 0.38$, $P < 0.001$ and $R^2 = 0.58$, $P < 0.001$, respectively; Fig. 7b). In addition to misregulation of mating responsive genes, there was a relatively large number of genes, particularly in *D. arizonae*, that were differentially regulated only in the heterospecific cross ("heterospecific only" genes) (Fig. 6). Expression changes of these genes in heterospecific crosses exhibited a strong positive correlation with expression changes in conspecific matings for all crosses and timepoints (Fig. 7a, b). Thus, although expression was significantly different between crosses, the differences were of relatively small magnitude.

GO-term enrichment analysis of DE genes in heterospecific crosses overlapped to some degree with conspecific crosses, particularly for *D. mojavensis* females (Fig. 5). However, several terms were enriched in one cross but not the other (Fig. 5), indicating that transcriptional disruptions in heterospecific crosses could have functional consequences. For example, in *D. mojavensis*, con- and heterospecific crosses showed significant enrichment for 12 and 11 categories GO-term categories respectively, from which only 5 categories overlapped between crosses (Fig. 5). In the case of *D. arizonae*, there were no overlapping categories between con- and heterospecific crosses (Fig. 5).

**Transcriptional divergence between species predicts transcriptional disruption in heterospecific crosses.** Postmating transcriptional divergence between the species was estimated for each gene as the absolute value of the difference between conspecific expression in *D. mojavensis* and *D. arizonae* (e.g., *D. mojavensis* conspecific expression divergence = |conspecific *D. arizonae* - conspecific *D. mojavensis*|). We then calculated the level of heterospecific disruption as the absolute value of the difference in expression values between hetero- vs conspecific transcriptional responses within each species (e.g. heterospecific disruption in *D. mojavensis* = |*D. mojavensis* heterospecific response – *D. mojavensis* conspecific response|). In *D. mojavensis*, the magnitude of expression disruption in heterospecific crosses was significantly positively correlated with expression divergence between the species (Fig. 8a). As the magnitude of expression disruption increased over time for conspecific mating responsive genes (Fig. 7a), the correlation between disruption and divergence strengthened as well ($R^2 = 0.14$, $P < 0.001$ and

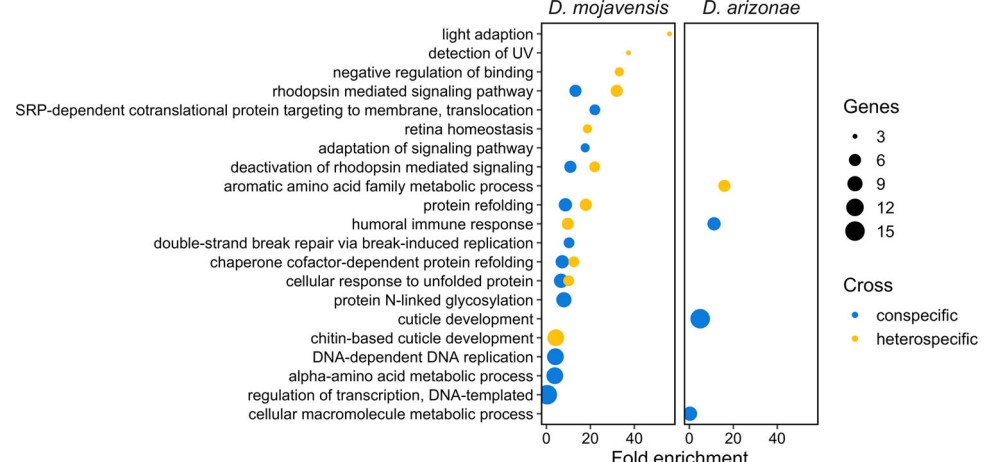

**Fig. 5 Gene ontology analysis of DE genes.** Functional analysis for significant DE genes, indicating distinct biological process gene ontology enrichment categories between the species for genes differentially regulated relative to virgins in con- and heterospecific crosses. The fold enrichment of detected genes within each enriched category is indicated.

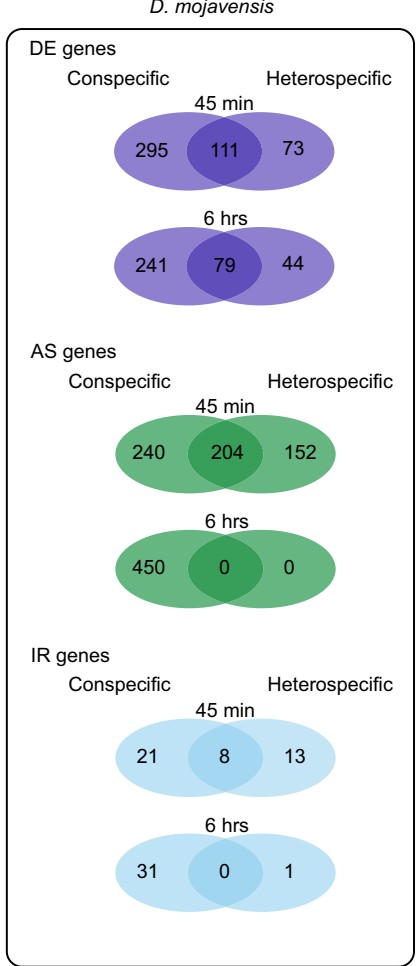
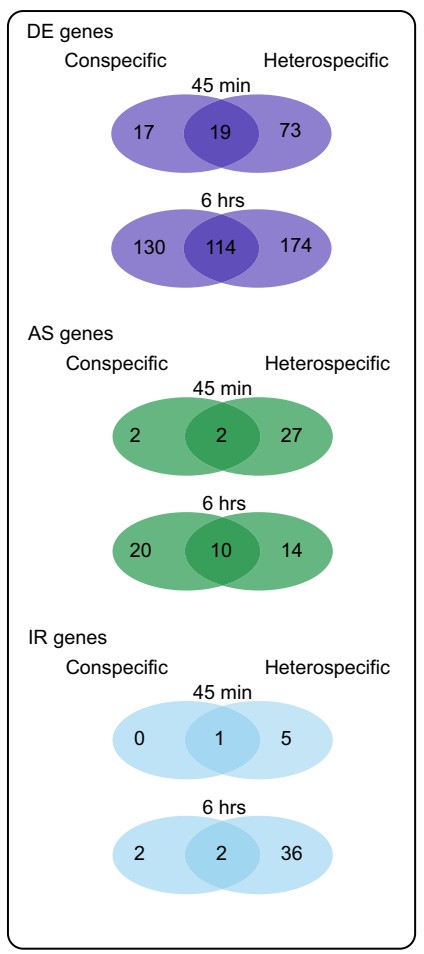

**Fig. 6 Comparison of differentially regulated genes in conspecific and heterospecific crosses in _D. mojavensis_ and _D. arizonae_.** Overlapping genes represent conspecific mating responsive genes that were properly regulated in heterospecific crosses. Genes unique to the conspecific cross represent conspecific mating responsive genes that were misregulated in heterospecific crosses. Genes unique to the heterospecific cross are those that are not part of the normal conspecific mating response but had disrupted transcriptional profiles in the heterospecific cross ("heterospecific only" genes).

$R^2 = 0.62$, $P < 0.001$, for 45 min and 6 h respectively; Fig. 8a). Likewise, there was a positive correlation between disruption and divergence in _D. arizonae_, though the strength of the correlation was more similar at both time points ($R^2 = 0.23$, $P < 0.001$ and $R^2 = 0.38$, $P < 0.001$, for 45 min and 6 h respectively; Fig. 8b). Altogether, these data indicate that expression of genes with more divergent postmating transcriptional profiles between species was more disrupted in heterospecific crosses.

**Molecular evolution of genes misregulated in heterospecific crosses**. In _D. mojavensis_, genes that were misregulated in heterospecific crosses (DE or AS), including conspecific mating responsive genes and those that were exclusive to the heterospecific cross ("heterospecific only" genes), exhibited lower ω values compared to the genome background in three out of four comparisons ($z = -3.86$, $P < 0.001$ for conspecific DE; $z = -8.74$, $P < 0.001$ and $z = -4.11$, $P < 0.001$ for con- and heterospecific AS, respectively; Fig. 9). In contrast, there were no significant differences between genes that were misregulated in heterospecific matings and the genome background in _D. arizonae_ (Fig. 9).

**Discussion**

A previous study demonstrated PMPZ barriers in crosses between all geographically distinct populations of _D. mojavensis_ females and _D. arizonae_ are strong[13]. Here, we demonstrate PMPZ

isolation in crosses between _D. arizonae_ females and _D. mojavensis_ males from the Mojave Desert (Fig. 1), which further underscores the importance of reproductive tract incompatibilities in this system and provides a comparative framework for examining the molecular basis of PMPZ incompatibilities. The reduction in fecundity we observed for _D. arizonae_ females mated to _D. mojavensis_ males is largely consistent with the previous study, which found similar reductions in oviposition by heterospecifically mated _D. mojavensis_ females from three of the four distinct populations[13]. Notably, the one exception to this pattern was for _D. mojavensis_ females from the Mojave Desert, which laid more eggs when mated to heterospecifics than conspecifics. The reduced fecundity we found in the reciprocal cross (Fig. 1a) suggests divergence in the mechanisms driving PMPZ isolation in the reciprocal crosses. Consistent with the previous study, we also observed a severe reduction in fertilization success and sperm viability in matings between _D. arizonae_ females and _D. mojavensis_ males (Fig. 1b–d). In fact, sperm storage appears to be more severely compromised in this cross relative to those involving _D. mojavensis_ females. At one day postmating the majority of heterospecific sperm in the seminal receptacle of _D. mojavensis_ females was motile, while the presence of motile sperm was sharply reduced at five days postmating. In contrast, in _D. arizonae_ we detected few motile heterospecific sperm even at one day postmating. This suggests the underlying causes of sperm

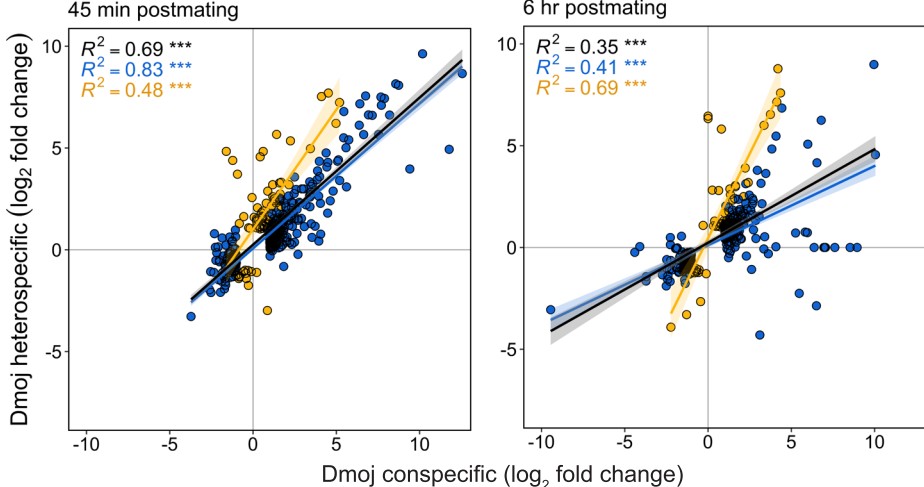

a. *D. mojavensis*

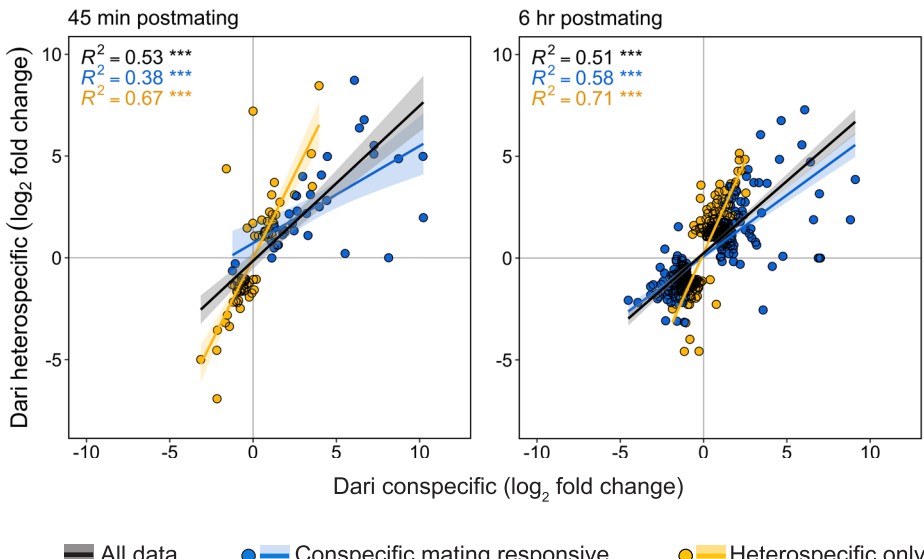

b. *D. arizonae*

**Fig. 7 Transcriptional correlations for DE genes between con- vs heterospecific matings in *D. mojavensis* and *D. arizonae*.** Scatterplots represent the relationship of relative fold changes (log₂) between con- vs heterospecific matings at 45 min and 6 h postmating in **a** *D. mojavensis* and **b** *D. arizonae*. Log₂ fold changes are relative to virgin females. Genes colored in blue are all conspecific mating responsive genes. Genes colored in yellow are those that were DE in the heterospecific cross but not the conspecific cross ("heterospecific only"). Pairwise Pearson's $R^2$ correlation coefficients and linear method trendlines (with 95% confidence intervals shaded) are shown. The black regression line is fitted through all the data (blue and yellow combined). $P$-values of correlations are noted: ***$P < 0.001$.

inviability may not be the same between the crosses. Additionally, other factors also likely contribute to the reduction in fertilization success, particularly in *D. mojavensis*, where females produce few fertilized eggs even during the time period where heterospecific sperm are still motile. Overall, these findings further establish the power of the *D. mojavensis*/*D. arizonae* study system for understanding the evolution and mechanistic basis of PMPZ isolation.

Postmating-prezygotic isolating barriers are often hypothesized to result from independent coevolutionary trajectories of sexual selection and sexual conflict in diverging populations. This leads to the prediction that species pairs with PMPZ isolation, such as *D. mojavensis* and *D. arizonae*, should exhibit divergent transcriptional responses to conspecific mating. As expected for recently diverged species, we found an overall positive correlation between the transcriptomic response to conspecific mating between the two species, which declined from 45 min to 6 h

postmating (Fig. 4). Nevertheless, these correlations were relatively weak, indicating substantial divergence between the species in the overall transcriptional response to mating. This conclusion is further supported by fundamental differences in transcriptional patterns of DE, AS, and functional categories associated with mating responsive genes in each species (Figs. 3, 5, and 6).

In general, *D. mojavensis* females displayed a higher number of DE genes in response to mating than *D. arizonae* females, especially at the 45 min postmating timepoint where nearly 11-fold more genes were DE in *D. mojavensis* females. This difference persisted at 6 h postmating though the magnitude of the difference was smaller (Fig. 3). While *D. mojavensis* showed a stronger early transcriptional response that declined over time, *D. arizonae* displayed the opposite pattern, with more DE genes detected at 6 h than at 45 min (Fig. 3). Interestingly, only ~5% of all mating responsive DE genes were differentially regulated in both species,

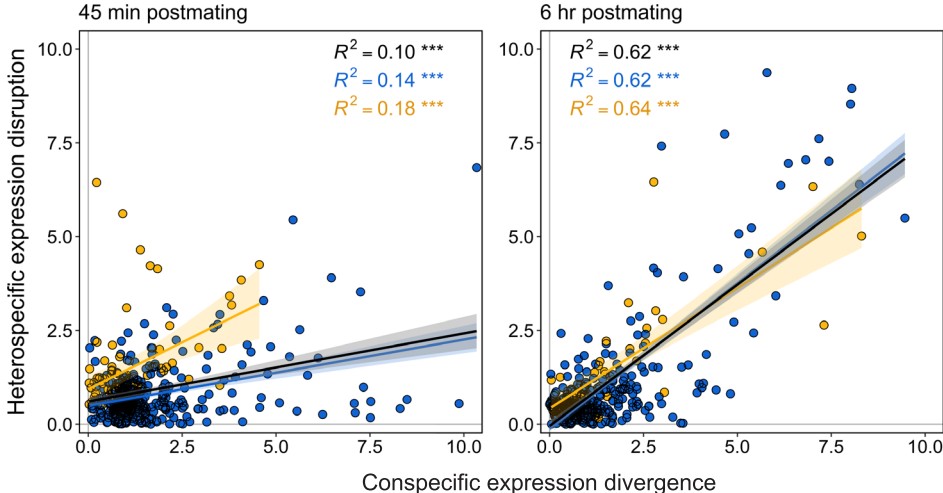

a. *D. mojavensis*

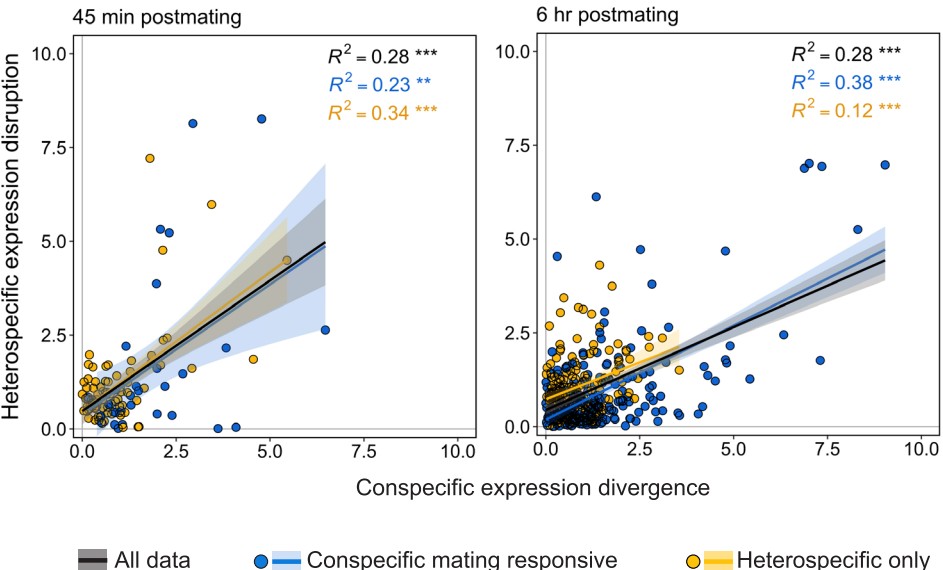

b. *D. arizonae*

**Fig. 8 Relationship between postmating conspecific expression divergence and the level of heterospecific disruption for conspecific-responsive DE genes.** Scatterplots show the relationship between postmating conspecific divergence (e.g., *D. mojavensis* conspecific expression divergence = |conspecific *D. arizonae*—conspecific *D. mojavensis*|) and the level of heterospecific disruption (*e.g.*, heterospecific disruption in *D. mojavensis* = |heterospecific *D. mojavensis*—conspecific *D. mojavensis*|) for **a** *D. mojavensis* and **b** *D. arizonae*. All DE genes detected in con- or heterospecific crosses at 45 min and 6 h postmating are included (FDR$_\alpha$ = 0.05). Genes colored in blue are all conspecific mating responsive genes. Genes colored in yellow are those that were DE in the heterospecific cross but not the conspecific cross ("heterospecific only"). Pairwise Pearson's $R^2$ correlation coefficients and linear method trend-lines (with 95% confidence intervals shaded) are shown. The black regression line is fitted through all the data (blue and yellow combined). *P*-values of correlations are noted: **$P < 0.01$, ***$P < 0.001$, ns not significant.

demonstrating striking divergence in the overall patterns of differential expression (Fig. 3b). Moreover, GO-term enrichment analysis of DE genes suggests potential functional divergence in the postmating response as we found no overlapping enriched categories between the species (Fig. 5).

Alternative splicing (AS) is known to be an important form of gene regulation in *Drosophila* female reproductive tissues[19], but, to our knowledge, has not been examined in the context of the postmating transcriptional response to mating. We found that AS is a key component of the female postmating response and that alternatively spliced genes are largely distinct from genes that are differentially expressed following mating (less than 5% of genes were found in both categories; Fig. 3a). Together with our recent

report of significant AS regulation in the head following mating in these species[20], our results suggest AS is an important but previously overlooked mechanism of the female transcriptomic response to mating. While most of the AS detected likely affects regulation of different protein isoforms, we also found differences in intron retention for many genes. Intron retention is a known mechanism of downregulation of gene expression as transcripts with retained introns trigger the nonsense mediated decay (NMD) pathway[21]. Consistent with this, we found intron retention was associated with transcriptional downregulation of genes following mating (Supplementary Fig. 1), which concurs with our previous findings of postmating transcriptional responses in the female head[20].

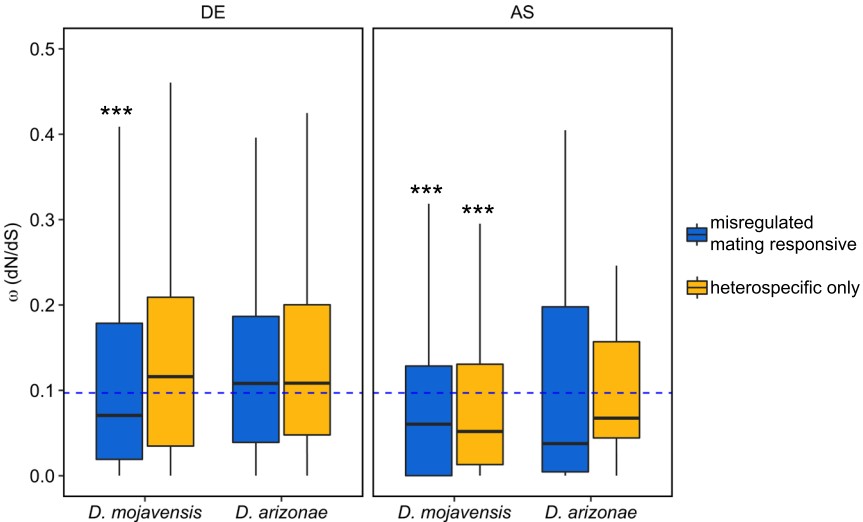

**Fig. 9 Median pairwise ω (dN/dS) for genes that were misregulated in heterospecific crosses.** Blue boxes represent mating responsive genes in conspecific crosses that were misregulated in heterospecific crosses. Yellow boxes represent genes that were differentially expressed only in heterospecific crosses. The median ω for the genome background (blue dash line) is indicated. Significant comparisons against the genome background using the Dunn method for joint ranking are indicated. *P*-values are noted: \*\*\**P* < 0.001. Boxplots represent the median with 25th and 75th percentiles, and whiskers show the 1.5 interquartile range.

Alternative splicing analysis also revealed substantial differences between species in the overall role of AS in the postmating response. Like the DE analysis, we found that AS was much more prominent in *D. mojavensis* than *D. arizonae* (Fig. 3). While AS was detected for hundreds of genes in *D. mojavensis* at both timepoints, few genes in *D. arizonae* were alternatively spliced in response to mating, especially at 45 min. Altogether, distinct patterns of DE, AS, and functional enrichment between *D. mojavensis* and *D. arizonae* demonstrate considerable divergence in the female transcriptomic response to mating despite relatively recent common ancestry. These results are consistent with a prior study showing divergence in the proteomic response to mating in *D. simulans* and *D. mauritiana*[6]. Together, these studies suggest that the female postmating molecular response evolves rapidly, which is consistent with predictions from evolutionary models of PMPZ isolation.

Divergence in the postmating response between species is expected to lead to incompatible interactions in heterospecific crosses. We found disruptions in the normal female postmating transcriptional response of both species following copulation with heterospecific males, including disruptions to the normal (conspecific) patterns of DE and AS (Fig. 6). Furthermore, several functional categories enriched following conspecific matings were not enriched in the heterospecific cross (Fig. 5), indicating transcriptional disruptions could have important functional consequences. While it is not clear whether these changes are causally linked to incompatibilities causing PMPZ isolation, the overall pattern supports the prediction that divergence in the conspecific postmating response results in considerable transcriptional disruption in heterospecifically-mated females.

Transcriptional disruption included misregulation of conspecific mating responsive genes and additional genes that were differentially regulated only in the heterospecific cross ("heterospecific only" genes) (Fig. 6). This pattern was also observed in a previous transcriptional analysis of heterospecific mating in *D. mojavensis*[7], and in a recent study of heterospecific matings between *D. novamexicana* and *D. americana*[8]. Misregulation of genes that normally respond to mating likely reflects disrupted interactions between male and female molecules in mismatched heterospecific crosses. While it makes sense that this could have adverse effects on reproductive outcomes, it is less clear to what extent the misregulation of non-mating responsive genes might also be involved in reproductive incompatibilities that give rise to PMPZ isolation. Differential regulation of these genes could represent transcriptional noise and be of little functional importance. However, if this were the case, we would expect an unpredictable pattern of misregulation. In contrast, these genes showed a consistent pattern where expression was generally in the same direction in conspecific and heterospecific crosses (Fig. 7). This pattern suggests that many of these "heterospecific only" genes are likely also responding to conspecific mating, albeit at a lower level.

Interestingly, Ahmed-Braimah et al.[8] found that a significant portion of genes that were differentially regulated only in heterospecific matings were linked to immunity, with heterospecific mating inducing a stronger immune response in females than conspecific mating. We found enrichment of differentially regulated immune genes (humoral response) in heterospecifically mated *D. mojavensis* females, while no enriched immune clusters were identified in conspecific matings. In *D. arizonae*, the same cluster of immune genes was enriched in conspecific, but not heterospecific matings (Fig. 5). Although these results could indicate an altered immune response following heterospecific copulation, this is not necessarily the case as enrichment only tests whether functional terms are overrepresented among DE genes. Since this depends on the total set of DE genes, which differs for each comparison, a cluster could be enriched in one comparison but not another even if the same genes were DE in each comparison. In fact, closer inspection revealed that almost all the genes in this functional cluster were DE in both conspecific and heterospecific crosses. The magnitude and direction of regulation was variable, but there were no patterns suggesting an overall higher immune response in conspecific or heterospecific crosses.

In general, the observation of differential regulation of immune related genes following mating is consistent with other studies in *Drosophila*, which have shown upregulation of immunity in response to mating[8,22–27]. While we did find that most differentially expressed immune genes, including antimicrobial peptides (AMPs), were upregulated following mating, there were a

few immune related genes that were downregulated. Most notably, *hemolectin* (*hml*), which is known to play a role in hemolymph clotting, was downregulated in all crosses. This is potentially relevant to the formation and/or degradation of the insemination reaction in the female reproductive tract of *D. mojavensis* and *D. arizonae* following mating. The function of the insemination reaction is not well understood, but some authors have hypothesized that it could be induced by males to prevent rapid female remating[17]. It superficially resembles a clot, and previous studies have identified several seminal fluid proteins in *D. mojavensis* that are known clotting factors[28]. While speculative, postmating regulation of female genes involved in clotting may play a role in the formation or persistence of the insemination reaction. We further note that although the insemination reaction is not efficiently degraded in heterospecifically mated *D. mojavensis* females[13], the severity and time-course of the insemination reaction has not been examined in heterospecifically mated *D. arizonae* females.

Although gene expression and alternative splicing were disrupted in both heterospecific crosses, patterns of disruption were largely specific to each cross. For example, the timing of and magnitude of transcriptional disruptions varied between the crosses, and the disrupted response in *D. arizonae* involved more genes that were only differentially regulated in the heterospecific cross compared to *D. mojavensis* (Fig. 6). Thus, despite similar outcomes of the heterospecific crosses, the underlying mechanisms causing reproductive incompatibilities are likely to be different. This finding is consistent with the prediction that sexual selection and sexual conflict follow divergent trajectories between isolated populations, targeting different traits in each population[11].

The standard model of the evolution of PMPZ isolating barriers posits that rapid evolution of traits involved in postcopulatory interactions results in reproductive mismatches between males and females from different populations. In a transcriptomic context, this leads to the prediction that genes with more divergent transcriptional profiles in conspecific crosses should also display more disrupted profiles in heterospecific crosses. We found an overall positive correlation between conspecific transcriptional divergence and heterospecific disruption (Fig. 8). This pattern was evident for conspecific mating responsive genes and for genes differentially regulated only in heterospecific crosses for both *D. mojavensis* and *D. arizonae*. Moreover, the strength of the correlation increased in both species from 45 min to 6 h hours postmating. While this overall pattern is consistent with predictions, there were some genes that showed similar patterns of regulation in conspecific crosses but had highly misregulated expression profiles in heterospecific crosses. This result is interesting because it indicates disruption of conserved aspects of the postmating response that could also play a role in incompatibilities.

Although misregulated mating responsive genes tended to be more transcriptionally divergent in comparisons of conspecific crosses, we did not find evidence that misregulated genes, as a group, evolve rapidly at the protein coding sequence level (Fig. 9). In fact, all categories of misregulated genes evolved at similar rates or, in some cases, below the genome median. This aligns with the results of our earlier study, which also found that misregulated genes in heterospecific crosses between *D. mojavensis* females and *D. arizonae* males did not evolve more rapidly than other female reproductive genes[10]. Ahmed-Braimah et al.[8] also reported a similar result in crosses between *D. americana* and *D. novamexicana*, suggesting that this might be a general pattern in *Drosophila*. One explanation for this finding could be that rapid changes in protein structure are likely to be most consequential for male and female proteins that directly interact, which is

consistent with previous studies showing rapid evolution of some female reproductive genes[27,29–33]. It is thus possible that many genes with disrupted expression profiles in heterospecific crosses are "downstream" of direct interactions between male and female molecules. As such, they may exhibit disrupted transcriptional profiles, while not evolving rapidly at the protein sequence level.

Overall, our data reveal that postmating transcriptional responses have diverged between *D. mojavensis* and *D. arizonae*, including striking changes in DE and AS that have potential functional implications. Moreover, we found that the normal transcriptomic response in each species was highly disrupted in heterospecifically mated females. Such disruption included misregulation of mating responsive genes as well as of additional genes that were differentially regulated only in the heterospecific cross. The patterns of disruption differed between the crosses, indicating potentially different mechanisms underlying reproductive incompatibilities. Importantly, mating responsive genes with more divergent transcriptional profiles in the two species generally displayed more significant disruption in heterospecific crosses, as would be expected if transcriptional disruption reflects failed or suboptimal interactions between the female reproductive tract and components of heterospecific ejaculates. While these findings are consistent with predictions of the standard model of the evolution of PMPZ isolation, we acknowledge that transcriptional disruption observed in heterospecific crosses may not be directly involved in incompatibilities that give rise to PMPZ isolation. Moreover, although transcriptional divergence between species and the level of transcriptional disruption in heterospecific crosses was positively correlated, the evolutionary forces that drove this divergence are not clear. Future research aimed at identifying specific male and female gene products that are directly involved in postmating incompatibilities is necessary to make causal links to PMPZ isolation. Identification of interacting genes will allow tests of whether these genes have evolved under divergent coevolutionary trajectories of sexual selection and sexual conflict.

## Methods

**Fly stocks.** All experiments were carried out using *D. mojavensis* and *D. arizonae* isofemale lines originally collected from Anza Borrego Desert State Park, Borrego Springs, CA (in 2002) and Guaymas, Sonora, Mexico (in 2000), respectively. The genomes of both lines have been sequenced and serve as reference sequences for mapping of RNA-seq reads (see below). Flies were held at 25 °C, under 12:12 h light:dark cycle and controlled density conditions in 8-dram glass vials with banana-molasses media[34] for all stocks and experiments.

**PMPZ isolation between *D. arizonae* females and *D. mojavensis* males.** Previous research has demonstrated strong PMPZ isolation in crosses between *D. mojavensis* females and *D. arizonae* males. To test for PMPZ isolation in the reciprocal cross, we compared fecundity, larval hatching, and fertilization success between con- and heterospecifically mated *D. arizonae* females. Fecundity was analyzed by pairing 8-12 day old virgin *D. arizonae* females with similarly aged *D. arizonae* or *D. mojavensis* virgin males in an 8-dram glass vial with banana-molasses media. Pairs observed copulating over a 24-hour window were kept and transferred to fresh vials every day for a week (10-11 pairs per cross). Oviposited eggs were counted, and fecundity was measured as the total number of oviposited eggs across the seven days.

The fecundity vials were used for the determination of hatching rate after 48 hours at 25 °C by counting total hatched larvae. Since reduced larval hatching could result from embryo inviability or lack of fertilization, we performed an additional experiment to help differentiate these possibilities. Virgin 8-12 old *D. arizonae* females were paired in vials with either a *D. arizonae* or *D. mojavensis* virgin male (8-12 days old). Copulations were observed ($N = 60$) and mated females were held together overnight in a large population cage to oviposit. In the morning, the food plate was removed and stored for 6 h to ensure any developing embryos would be 6 to 22 h old. Embryonic development time for these species is approximately 28 hours[35]. Eggs were dechorionated in 2.5% sodium hypochlorite before embryos were fixed in a of 1:4 solution of 4% paraformaldehyde/heptane and then devitellinized in methanol. Embryos were stained with 4′,6-diamidino-2-phenylindole (DAPI; 2.8 µg/ml) and analyzed by fluorescence microscopy using a Leica DM5000B microscope (20X objective) to determine whether embryonic

development had been initiated. Given our methodology, the minimum age of embryos was 6 hours, a point at which developing embryos are clearly identifiable. However, embryogenesis could have been arrested at earlier stages of development if embryos were inviable. Although we considered eggs to be unfertilized if there were no more than four nuclei observed (representing four products of female meiosis), we acknowledge that if embryogenesis was arrested during the earliest syncytial divisions, these embryos would be difficult to distinguish from unfertilized eggs.

To evaluate sperm storage, we paired virgin *D. arizonae* and *D. mojavensis* females with conspecific or heterospecific males as described for the fecundity assay. Following successful copulation, males were removed, and females were pooled in vials (2-10 females per vials). Females were maintained at 25 °C for either 24 h or five days before dissection. Female lower reproductive tracts (LRTs) were dissected in sperm buffer (0.05 M Tris, 1.1% NaCl, 0.1% dextrose, 0.01% L-arginine, 0.01% L-lysine), placed on a slide, covered with a glass coverslip, and visualized on an Olympus inverted light microscope. A total of 110 dissections were performed (10–17 per species-cross-age). Sperm motility within the seminal receptacle was classified by eye into one of four categories: many (wave-like sperm tail activity observed across nearly the entirety of the seminal receptacle); medium (one or two regions of wave-like sperm tail motility); few (tens or fewer visible motile sperm scattered along the seminal receptacle lumen); none (no motile sperm).

**LRT sample collection, RNA extraction, cDNA library construction, and sequencing**. Our experimental design involved transcriptomic analysis of conspecific and heterospecific matings (Fig. 2). All mating experiments were performed using virgin flies (8–12 days old) by isolating pairs in vials and observing copulation events during a 2-h window in the morning. Males were removed from vials after copulation and females were kept in the vials until the specified postmating period was reached (45 min or 6 h), when LRTs were collected (20 per sample) from both mated and virgin females to generate RNA-seq libraries. Postmating times represent early, and later major transcriptional changes based on results from Bono et al.[7] in con- and heterospecifically-mated *D. mojavensis* females. Three replicates were collected per experimental cross, which generated a total of 30 samples (Fig. 2). All tissues were placed immediately in TRIzol and kept at -80 °C until total RNA extractions. Total RNA was extracted using Direct-zol RNA kit (Zymo Research). Both RNA quality and quantity were inspected on a Bioanalyzer (Applied Biosystems/Ambion). cDNA libraries were generated using KAPA Stranded mRNA-Seq Kit following manufacturer's instructions. Libraries were sequenced at Novogene Inc. using the HiSeq SBS v4 High Output Kit on Illumina platform flow cells with runs of 2 ×150 bp paired-end reads. Illumina's HiSeq Control Software and CASAVA software (Illumina, Inc.) were used for base calling and sample demultiplexing.

**Sequence trimming and mapping**. Nearly 700 million total paired-end read sequences were obtained from the Illumina runs, ranging from 11 to 34 million raw paired-end reads for each sample. Reads were trimmed for quality, and adapter sequences were removed using a minimum quality base of Q = 20 and minimum read length of 50 bp using the software Trimmomatic[36]. Trimmed reads were then mapped to corresponding reference genomes using splice-aware mapper GSNAP[37] with the option of new splice events detection. Template based genomes (both containing 13,211 coding genes) from the same *D. mojavensis* and *D. arizonae* lines utilized in this study were used for mapping RNA-seq reads. For *D. mojavensis*, the assembly from[38] (Accession number SRP190536) was used with updated annotations retrieved from FlyBase version FB2016_05[39]. A template genome version of *D. arizonae* (osf.io/ukexv) was assembled using our previous method for *D. mojavensis*[38], with paired-end and mate pair Illumina reads from[40] (Accession number SRP278895). Generated *sam* files were converted to *bam* format after indexing and filtering for a minimum mapping quality of MQ = 20 using SAMtools[41]. These mapping results were then used for all differential expression and alternative splicing downstream pipelines. Previous evidence indicated the transfer of male RNA transcripts to females during copulation in crosses between *D. arizonae* males and *D. mojavensis* females[7]. Therefore, to ensure that our transcriptional analysis corresponded to the female transcriptional response, we removed all genes whose transcripts were completely male-derived from subsequent analyses. Any transcripts with sequencing reads from both males and females were retained in the dataset.

**Differential expression (DE)**. We created a gene level read count matrix for all samples using *featureCounts*[42]. The read count matrix was filtered for a minimum count cutoff of 3 cpm in at least two out of three replicates per group. All DE analyses were performed using the R package *edgeR*[43] after *TMM* library normalization. All comparisons were performed between mated females (con- or heterospecific mating) and virgin females (*D. mojavensis* or *D. arizonae*) at each postmating period (45 min and 6 h).

**Alternative splicing (AS) and intron retention (IR)**. We used *JunctionSeq*[44] to detect genome-wide patterns of AS genes. The *JunctionSeq*[44] pipeline is based on differential usage calculated from both exon and junction feature coverages. A new

flattened *GTF* annotation file that excluded overlapping features was first generated using *QoRTs*[45]. All overlapping genes were merged as composed by a flat set of non-overlapping exons and splice junctions with unique identifiers. *QoRTs* was also used to generate a read count matrix for AS analysis, including three types of read counts per gene as estimated by exons, junction, and gene level counts. No read was counted more than once in the model since exon and junction dispersions are fitted independently.

Intron retention (IR) is a specific type of AS that is not necessarily captured by *JunctionSeq* and is generally considered as a mechanism of expression downregulation, as transcripts with retained introns are degraded by the nonsense-mediated decay pathway[21]. However, in some cases transcripts with retained introns have been shown to perform novel functions[46–48]. We investigated whether postmating AS events also involve IR using the *IRFinder* pipeline[49]. For each genome, a new reference annotation was built by removing all overlapping intron features then unique identifiers were assigned to each flattened exon. Only regions with high mapping scores as estimated through simulated reads across the genome were included in the flattened annotation file. A read count matrix with all reads overlapping splice junctions was generated and IR rates were estimated as junction reads / (junction reads + intronic reads) for each sample.

**Postmating transcriptional divergence and disruption of gene expression in heterospecific matings**. Analysis of differential gene regulation in conspecific crosses established the typical postmating transcriptional response in each species. We refer to genes that were differentially regulated in response to mating as "conspecific mating responsive genes". To determine the extent of postmating transcriptional divergence between *D. mojavensis* and *D. arizonae*, we compared conspecific mating responsive genes identified in both species. Genes differentially regulated in both species represent the conserved postmating transcriptional response, while genes regulated in a species-specific manner indicate divergence in the postmating response. Four genes from the six-hour comparison were differentially regulated in both species but in opposite directions. While these genes are responsive to mating in both species, they were categorized as part of the species-specific response since the direction of regulation was not conserved. To quantitatively compare the overall postmating response in both species we computed correlations between log₂ fold changes relative to virgins in conspecifically-mated *D. mojavensis* and *D. arizonae* females. This analysis was only conducted for DE genes given the low number of AS/IR genes in *D. arizonae*. We included all conspecific mating responsive genes in both species. The strength of the correlation indicates the degree of transcriptional divergence, with a weaker correlation between the crosses indicating more divergence.

To examine whether the normal conspecific mating response was disrupted in heterospecific crosses we compared differentially regulated genes (DE, AS, and IR) between conspecific and heterospecific crosses within each species. Genes differentially regulated in both crosses are conspecific mating responsive genes that were properly regulated in the heterospecific cross. Genes exclusive to the conspecific cross are those mating responsive genes that were misregulated in heterospecific crosses. Genes exclusive to the heterospecific cross represent those that are not part of the normal conspecific mating response but had disrupted transcriptional profiles in the heterospecific cross ("heterospecific only" genes). As above, one gene from the six-hour comparison that was differentially regulated in both crosses but in opposite directions was categorized as a misregulated conspecific mating responsive gene. To further compare the overall magnitude of transcriptional disruption in heterospecific crosses, we examined correlations between log₂ fold changes in conspecifically- and heterospecifically-mated females relative to virgins in each species. This analysis was only conducted for DE genes given the low number of AS/IR genes in *D. arizonae*. We computed correlations using conspecific mating responsive genes and genes that were differentially regulated in heterospecific but not conspecific crosses ("heterospecific only" genes). A weaker correlation indicates more transcriptional disruption.

We then investigated whether the level of heterospecific expression disruption is predicted by the amount of transcriptional divergence between the species. Postmating transcriptional divergence between the species was estimated as the absolute value of the Euclidean distance of conspecific relative expression values (fold change relative to virgin) for each gene, for example:

$$D.\ mojavensis\ \text{conspecific expression divergence} = |\text{conspecific } D.\ arizonae - \text{conspecific } D.\ mojavensis|$$

(1)

We estimated the level of heterospecific disruption as the absolute value of the Euclidian distance of relative expression values (fold change relative to virgin) between hetero- vs conspecific transcriptional responses, for example:

$$\text{heterospecific disruption in } D.\ mojavensis = |D.\ mojavensis\ \text{heterospecific response} - D.\ mojavensis\ \text{conspecific response}|$$

(2)

We used the absolute value because our hypothesis was that transcriptional divergence would be positively correlated with the amount of transcriptional disruption irrespective of the direction of transcriptional changes. This analysis was only conducted for DE genes given the low number of AS genes in *D. arizonae*. Correlations for each species were computed using conspecific mating responsive genes and genes differentially expressed in heterospecific but not conspecific crosses ("heterospecific only" genes).

**Functional and molecular evolutionary analyses.** Overrepresentation of specific categories of biological process were then investigated for DE genes using *Panther*[50]. We did not conduct this analysis for AS/IR genes due to the low number of such genes identified in *D. arizonae*. Additionally, since genes evolving rapidly due to sexual selection and sexual conflict are predicted to be involved in PMPZ isolation, we investigated signatures of positive selection on the set of DE and AS genes that were misregulated in heterospecific crosses. This included misregulated conspecific mating responsive genes, and genes that were differentially regulated in heterospecific but not conspecific crosses ("heterospecific only" genes). We estimated rates of molecular evolution ($\omega = d_n/d_s$) using *codeml*, part of *PAML* 4.9[51]. CDS alignments between *D. mojavensis* and *D. arizonae* were produced with *MUSCLE* 3.8.31[52]. Any alignments with internal stop codons or frameshifts were removed before analysis. Furthermore, raw synonymous and nonsynonymous polymorphism counts were generated with *KaKs* Calculator 1.2[53] and loci lacking synonymous substitutions were omitted from the analysis. *Codeml* was run using model *0* with default values. Pooled gene lists from both time points were used for all statistical analyses. Higher $\omega$ values for misregulated genes compared to the genome would be consistent with positive selection or relaxed constraint driving rapid evolutionary change in these genes.

**Statistics and Reproducibility.** For the fecundity assay, statistical differences in egg counts were assessed using a *t*-test after verifying that the residuals were normally distributed and variances were homogenous. Hatching rate of oviposited eggs were analyzed with the *afex* package[54] in R using a generalized linear mixed model (GLMM) with binomial error term and logit link function. Female identity was treated as a random variable to account for the fact that multiple measurements were taken from a single female. The DAPI data on egg fertilization was analyzed in R using a generalized linear model (GLM) with binomial error term and logit link function. The sperm motility categorical data was analyzed as ordinal variables using an ordinal logistic regression in JMP v16.

For DE analyses, normalized counts were analyzed by a GLM using a negative binomial model. To identify DE genes, a false discovery rate (FDR) correction following a global α of 0.05 was used for multiple comparisons[55], as well as a log$_2$-fold-change threshold of 1.0. For AS analysis by *Junctionseq*, differentially alternatively spliced genes identified when at least one exon or splice junction was differentially used (relative to overall expression of the gene) between mated females (con- or heterospecific mating) and virgin females (*D. mojavensis* or *D. arizonae*) at each postmating period (45 min and 6 h). A FDR correction following a global α of 0.01 was used for multiple comparisons[55], as well as a log$_2$-fold-change threshold of 1.0. For intron retention, the generated count matrix was used by *IRFinder* R package to estimate the GLM using the *DESeq2*[56] R package framework. A FDR correction following a global α of 0.05 was used for multiple comparisons[55], as well as a log$_2$-fold-change threshold of 1.0. Because IR changes could serve as a mechanism of downregulation by transcript degradation, we tested this hypothesis by estimating IR changes for up vs. downregulated genes between mated vs virgin samples (IR change). A GLM analysis was performed using categories of up and down regulation as independent variables and the level of IR change as the dependent variable for each mating experiment. GLM analysis was performed using a gaussian family distribution with identity link function after square root transformation of data to ensure assumptions of the GLM were met.

All correlation analyses were conducted using R. Pairwise Pearson's $R^2$ correlation coefficients and linear method trend-lines (with 95% confidence intervals shaded) were generated. Comparisons of molecular evolution were performed using JMP v16. For each species and cross type, significant deviation from the genome-wide $\omega$ was determined using the Dunn method for joint ranking.

**Reporting summary.** Further information on research design is available in the Nature Research Reporting Summary linked to this article.

## Data availability

All reads have been deposited at NCBI under BioProject ID PRJNA777940. All other data has been deposited in the Supplementary Data 1 and 2.

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

## Acknowledgements

We would like to thank undergraduate students Nathaniel Talamantes, Rorie Shae Robinson, Kamal Jitendra Patel, Moruj Athma and Graham Wegner from the University of Arizona and Dr. Helen Pigage at UCCS for assistance in collecting the *Drosophila* samples. The research presented in this publication was supported by the Eunice Kennedy Shriver National Institute of Child Health and Human Development of the National Institutes of Health under award number R21HD097545 to LMM and JMB as well as by funds from the University of Arizona to LMM.

## Author contributions

L.M.M. and J.M.B. conceived the initial idea and the project design. F.D. performed most of the fly work, library construction and transcriptomic analyses. C.W.A., J.M.C., X.C., L.M.M., and J.M.B. participated in fly work. All authors, but mainly F.D., J.M.B. and L.M.M., participated in the data analysis and the writing of the manuscript. All authors read and approved the final manuscript.

## Competing interests

Luciano Matzkin is an Editorial Board Member for *Communications Biology*, but was not involved in the editorial review of, nor the decision to publish this article. All other authors declare no competing interests.
