## [Peer Review File · Communications Biology]

Reviewers' comments:

Reviewer #2 (Remarks to the Author):

The authors addressed all my concerns in this revised version of the manuscript.

Reviewer #4 (Remarks to the Author):

First, I found the paper difficult to read and (like Reviewer 3) had to reread many sentences and whole paragraphs to understand what the authors were trying to say. I do not think the authors took this comment as a major concern. They say that "have gone through the entire text and when appropriate have simplified the language" (underlining is my emphasis). I do not know what the first version looked like, but the revision is still hard to follow. I suspect that they did not find it "appropriate" to simplify the language in many places.

Also regarding "major concern 1", I agree with Reviewer 3 that the results are hard to follow. The authors say they added a new figure 1 outlining the experimental design, but this figure does not help to reduce the cognitive load. I do not think this figure is necessary because this aspect of the experimental design is easy to follow. The complexity that is hard to follow (and that I imagine Reviewer 3 is referring to) is keeping track of which subsets of DE (or AS) genes are being analyzed. For example, in Figure 3, what are the conspecific only, heterospecific only, and con- and heterospecific columns referring to? Do these refer to the different sets from the Venn diagram on the right? Is it appropriate to lump them together in a single column if the DE and AS lists come from different datasets? The figure legend does not help. Panel A does not simply report the number of DE, AS, and IR genes following con- and heterospecific matings. (That figure would just have two columns: one for the conspecific matings and one for the heterospecific matings.) These data have been subsetted further, but it's hard to follow exactly how.

Similarly, I found all of the scatterplots hard to follow. How are Figures 4, 7, and 8 substantially different from each other and how were the blue and yellow subsets identified? As recommended by the reviewer, it would be helpful to have some figure explaining how the conspecific mating responses versus heterospecific only genes were identified. For conspecific expression divergence and heterospecific disruption, is it appropriate to calculate these using absolute values? Seems like you're losing potentially important information by forcing all of these comparisons to be positive.

Regarding "major concern 2", I think there may still be remnants of the old model. On line 62, I do not know what "this model" is referring to. Perhaps the coevolutionary divergence model that has now been (mostly) deleted? I agree with Reviewer 3 that the co-evolutionary divergence model is not unique and doesn't need a special name.

Line 179: the authors did not address the concern about DE comparisons. This aspect of the study is still very difficult to follow (see comment above).

Response to reviewers

We would like to thank the editor for considering our manuscript for publishing in *Communications Biology*, as well as the reviewers for their valuable suggestions. We have carefully read all suggestions and questions raised by the reviewers, which have helped us to improve the revised version of our manuscript. Here we explain how we responded to these suggestions point by point.

Reviewer #1 (Remarks to the Author):

This is a carefully done and clearly presented study that looks at whether and how the transcriptional activity of the female reproductive tract changes upon mating. It asks whether those changes are different for conspecific versus heterospecific mating, whether any changes occur in both directions of heterospecific matings, and if there are any patterns to the transcriptional changes when comparing conspecific and heterospecific matings. The authors use a well-established study system of a *Drosophila* species pair, a solid and well-justified experimental design, thorough analysis, convincing interpretations of their findings, and discussion of alternative interpretations and avenues for further study. I find this a very nice contribution to the literature, offering up important new datasets and biological insights, and have only minor suggestions for potential improvement.

R// We thank the reviewer for the positive considerations about our manuscript.

Major Concerns:

None.

Minor Concerns:

1. line 47: I would remove the word “other” since gametes are not generally considered molecules.

R// Agreed and changed.

2. line 105: what is the “insemination reaction”? the context suggests it is a physical object rather than a chemical reaction. Please briefly explain for readers unfamiliar with this term.

R// Agreed. The term “insemination reaction” has now been explained (Lines 91-95).

3. line 142: please justify use of a t-test here: did the authors ensure the distributions they were comparing were normal?

R// Agreed. We tested for the normal distribution and homoscedasticity of the data as required for the t-test. This is now indicated in methods (Lines 132-134)

4. line 152: (a) what is the normal timing of embryonic development through to hatching in these species? Adding this information will help readers understand why the 6-22 hour window was an appropriate window to choose for assessment of fertilization. (b) please add additional methodological details about the staining procedure. Were eggs dechorionated, fixed, and devitellinized? What concentration of DAPI and what sort of microscopy were used? Although this technique is very routine in *Drosophila melanogaster*, adding these methodological details will help readers appreciate whether it was applied to these species in such a way that would indeed allow unambiguous distinction between fertilized and unfertilized eggs.

R// Agreed. We have added more information on the DAPI staining methods (Lines 146-156).

5. line 160: (a) suggest moving the word “either” to go between “for” and “24 hrs” (b) I believe this is the first place where the abbreviation LRT is used, but it is not defined until line 175 - please provide definition at the first instance of acronym use.

R// Agreed. We have now clarified the LRT term in the suggested place and corrected the typing mistake (Lines 160-161).

6. line 164: was this classification done qualitatively by eye or with some kind of quantitative measurement? Please clarify this method.

R// Agreed. This classification was performed by eye, based on major differences in sperm motility. The method has now been indicated in the corresponding section (Lines 165-166).

7. line 321: which libraries are being discussed in this section? ones derived from the female *D. arizonae* x male *D. mojavensis* cross described in the section immediately preceding this one, or others? Please add a sentence or two of context to the beginning of this section for clarity.

R// Agreed. In this section of results, we refer to the RNA-seq libraries derived from LRT following con- and heterospecific crosses between *female D. arizonae* x *male D. mojavensis*. We have made changes to the text for clarification of the discussed libraries and referenced to a new figure showing the experimental design used for these analyses (Lines 331-336, and new Fig 1).

8. line 380: change “form” to “from”.

R// Agreed and changed.

9. Figure 5: please define “DE” and “AS” in legend for clarity.

R// Agreed. DE and AS definitions have now been included in the legend of Figure 5 (now Fig 6).

10. lines 470-471: which four cactus host populations? Although a citation is provided here, it

would help the reader to briefly explain what these four populations are and how they relate to the experimental design in this work.

R// Agreed. We now introduce the four geographically/ecologically isolated populations of *D. mojavensis* in the introduction (Lines 95-99). We removed references to cactus hosts as this is not relevant to the current study.

Reviewer #2 (Remarks to the Author):

The manuscript by Diaz et al. examines the transcriptional response of *Drosophila* females following con- or heterospecific mating. The study system used is an excellent example of PMPZ isolation, and the authors utilize this system to understand specific characteristics of the postmating response given the conceptual expectations within this PMPZ system, and frame the study within the predictions of coevolutionary divergence model. The study is executed well, the paper is very well written, the conclusions are sound, and the results are an important advance in our understanding of systems level processes related to PMPZ, speciation, and the divergence in gene expression processes.

R// We thank the reviewer for the positive comments about our manuscript.

I have one concern about the analysis approach that I hope the authors can address, namely that the conclusions are not influenced by differences between the quality of the two species' genome/annotation. Specifically, one major result in the paper is the high discrepancy in conspecific DE numbers between the two species, especially in the earlier timepoint. This result is stunning if it is not a technical artifact of the upstream analysis process. The authors should be able to persuade the reader that this is not a concern by adding a few summary statistics of the mapping results and additional information about genome annotation (see comments below). I also have a few basic specific suggestions below.

R// We thank the reviewer for pointing us to this important consideration as we failed to explain this point in our results section. We have now included this explanation in our results (Lines 331-336) and included a new supplementary figure (new Fig S1) displaying obtained library sizes and mapping statistics per genome. As indicated in this section, the new figure shows that the mapping rates are consistently similar in both species' genomes with ~ 83% of mapped reads. This mapping rate is quite consistent across mating experiments and time points in both species (new Fig S1). Additionally, we indicated that the template-assembled genomes used for mapping had the same number of transferred annotations for coding genes, 13,211 (Lines 204-207).

- Methods:

o Lines 152-155: This assay doesn't exclude the possibility of early onset defects at the initiation of syncytial divisions, but not a big concern given the other assays. But the authors should address whether this assay has short-comings with respect to differentiating fertilization from early cytoplasmic incompatibility.

R// Agreed. This is a good point. We now acknowledge this caveat in the methods (Lines 152-156) and the results (Lines 316-323). In the results we also describe why we think the data are most consistent with failed fertilization as opposed to early embryonic inviability.

o One concern with the approach in this study is the potential discrepancy in genome quality/annotation between the *mojavensis* genome (very good quality) and the *arizonae* genome (not as good?). More detail is needed here for the reader to confidently conclude that the *arizonae* template genome is not deficient in terms of its gene annotation. For example, the authors should report read-mapping statistics in the supplement. I raise this point because the wide discrepancy in the number of DE genes between the two species could be explained by failed mapping consistency between the two datasets. I suspect this isn't a major concern, but the authors should address this, even if just in the supplement. Also, since *moj* and *ari* are so closely related, have the authors tried mapping each species' reads to the other species to examine mapping discrepancies? This can often show if read mapping is reduced between species due to divergence or due to genome quality. (Just a thought, not a proposition to include these results).

R// We thank the reviewer for calling our attention to this important aspect of genome divergence between species. A new supplementary figure showing mapping statistics has now been included (new Fig S1). The new figure shows consistently similar mapping rates in both species' genomes (~ 83%). Please, see a more detailed response to this concern above.

- Results:

o Line 308: Remind the reader here how the *D. mojavensis* female phenotypic results from previous work compare to the reciprocal cross.

R// Agreed. A new section was included introducing previous PMPZ results involving *D. mojavensis* females (Lines 306-310).

o The statement about where the reads are deposited shouldn't be here.

R// Agreed. We removed the statement on data availability from the results section as this is already indicated in a separate section (Data availability, lines 705-708).

o Lines 349-360: Refer to my comments in the Methods about read mapping discrepancy—the results reported here could be driven by some artifactual aspect of the two species' read-

mapping properties. I hope it's not, because these are really striking results, but be sure to address alternative issues with this result outcome (it came to my mind right away).

R// We thank the reviewer for this clarification. Please, see a more detailed response to this concern above.

o Lines 382-385: The statement that “transcription disruptions in heterospecific crosses could have functional consequences” needs more interpretation here. Specifically, how does one draw that conclusion from Figure 4? I suspect the authors are hoping the reader can reach that conclusion from seeing that some GO categories are only enriched among the heterospecific category, but it would help for the authors to spell that out for the reader.

R// Agreed. The statement about the functional consequences of transcriptional disruption in the heterospecific crosses has been more clearly explained based on the insights in Figure 4 (now Figure 5, Lines 404-411)

- Figures:

o Figure S1 has typo: “..in of IR..”

R// Agreed. The typing error has been corrected in the legend of Figure S1 (now Fig S2).

Reviewer #3 (Remarks to the Author):

Gene expression in the female reproductive tract changes after mating, often in ways that are thought to be in direct response to components of the male ejaculate. Interaction of male and female reproductive proteins both come from and contribute to male-female coevolution. Here, Diaz and colleagues propose that this male-female coevolution, when it occurs in allopatry, can lead to divergence in reproductive compatibility that promotes post-mating, prezygotic (PMPZ) isolation. They refer to this process as the coevolutionary divergence model. This study seeks to test this model by characterizing how disruption of the post-mating gene expression response differs in reciprocal directions of a heterospecific cross between sister species *D. mojavensis* (Dmoj) and *D. arizonae* (Dari). In so doing, they build on past work by adding the Dari female x Dmoj male cross and specifically looking for differences in alternative splicing, including intron retention. The expectation is that there will be substantial differences in genes that comprise the post-mating gene expression response for each species and the ways in which this response is disrupted in heterospecific matings will also be different.

This study combines many of my favorite things – female reproductive tracts, gene expression, and speciation. It is an important piece of the PMPZ puzzle, because it includes both directions

of a reciprocal cross and also looks specifically at patterns of alternative splicing, which are understudied within reproductive contexts. However, there are a number of aspects of the paper that do give me pause.

Major concerns

1) First, it is not as easy to read as I would like it to be. I often found myself rereading sentences and whole paragraphs to decipher the message. Part of the issue was overly long sentences, overly complex language, and burying the main point in both of these.

Another part of the issue is that the study itself is not simple and so really requires clear communication to explain it well. The experimental design is somewhat complex for the average reader accustomed to simpler DE analyses, because the result of interest is more than a list of DE genes. Rather, it's about if and how the subset of DE genes changes between conspecific and heterospecific matings, and whether that change differs depending on direction of the reciprocal cross. Because of unclear writing (and also some figures), it was often difficult for me to tell what was done, why, what was found, and why it is important.

R// Agreed. We have added a new figure 1 outlining the experimental design of this study. Additionally, we have gone through the entire text and when appropriate have simplified the language.

2) Second, I struggled to distinguish the coevolutionary divergence model from plain old allopatry. Allopatric speciation works, because populations evolve independently in isolation from one another, reproductive and/or developmental regulatory pathways diverge leading to incompatibilities in mates or hybrids, and reproductive isolation is born. It is generally understood that drift is not enough to generate such significant differences; they must come from selection, and that selection is often inflicted on a population by other organisms, and often in a coevolutionary manner. So do we really need a coevolutionary divergence model? (As a side note, I suggest reversing the order of the name from coevolutionary divergence to divergent coevolution, since it is the coevolution that is divergent, rather than the divergence that is coevolutionary. I had a hard time understanding what the model was about until I switched the order of the terms.) If we were to stick with a separate model for the role of coevolution in divergence, I would expect to test it by comparing genes that are known to be coevolving with genes that are not. Or perhaps comparing post-mating gene expression changes in female reproductive tract vs. head, to get at reproductive vs non-reproductive processes. Both of these options are not easy to get it at – the first, because we still don't know much about which male and female proteins are interacting with each other after mating, and the second, because that would require a whole other big study. So in short, I am not convinced that this study really tests this coevolutionary divergence model, but I also don't think it needs

to. I think it is enough to test the hypothesis that independent evolution of Dmoj and Dari would result in gene expression disruptions after heterospecific matings that are themselves species-specific.

R// The divergent coevolution model we discussed is a more specific allopatric speciation model that has been proposed to explain the evolution of PMPZ isolation. The main point is that divergent coevolutionary trajectories of sexual selection and sexual conflict drive the evolution of incompatibilities, which would not necessarily be the case for a more general allopatric speciation model. Given the good points made by the reviewer, we have deemphasized the model in the revision. While we do discuss the idea that sexual selection and sexual conflict are predicted to drive PMPZ isolation, we no longer use the model and its predictions to organize the paper.

3) Third, there isn't much we can say about mechanisms from this study. This third point may not be an issue if Comms Bio is okay with it. I realize that getting at function and mechanisms is a tall ask for non-model *Drosophila*, and the dataset as it stands, in my opinion, makes a substantial contribution to the field. That said, perhaps more could be speculated about certain genes in the Discussion. I would be okay with some bit of hand-waving, so long as the speculation is rooted in evidence.

R// While we appreciate the reviewer's comment, we are concerned that speculating more about the role of individual genes will detract from the main message about comparative patterns of gene regulation in the different crosses. As pointed out by the reviewer, the interpretations of the data are already complex, and we are concerned about adding anything more to further muddy the waters.

Specific comments:

Commas: I'm a bit of a stickler about these, because they can really help clarify a sentence.

L43, after "historically".

L56, after "frequently".

L190, after "quality".

L301, after "(Fig. 1a)".

R// Agreed. We changed all but the first suggestion (historically) because in that case the comma should follow the dependent clause.

L57-58: Change "Independent coevolutionary dynamics between males and females" to "Male-female coevolutionary dynamics".

R// Agreed and changed (Line 55).

L60: Add something like “We refer to this hypothesis as the divergent coevolution model.” – if you choose to keep the model in the paper.

R// See comments above about the model.

L64: Change “of” to “due to”.

R// Agreed and changed.

L62-67: An overall summary of the model is needed to place the predictions in context

R// This passage has been removed based on the reviewer’s comments about the model.

L67: Change “Postmating molecular interactions following conspecific mating” to something like “Post-mating interactions between male- and female-derived molecules”.

R// This passage has been removed based on the reviewer’s comments about the model.

L68: Delete “as a result of different coevolutionary trajectories”.

R// This passage has been removed based on the reviewer’s comments about the model.

L68: Edit to “(2) This divergence will cause postmating molecular interactions to be disrupted...”.

R// This passage has been removed based on the reviewer’s comments about the model.

L68-71: Prediction 2 is not specific. Is the coevolutionary divergence model predicted to explain PMPZ isolation, or is it postulated as one of multiple possible mechanisms of PMPZ isolation? It isn’t clear what this model specifically is postulated to explain. It seems to be hypothesizing that speciation occurs because of independent evolutionary trajectories of two lineages, which seems to be the same as allopatry. I’m having a hard time seeing how this model differs from other models of speciation and how it fits in with PMPZ isolation.

R// See comments above about the model.

L67-76: I would expect that genes whose products are known to be involved in male-female interactions are also more likely to be dysregulated in heterospecific matings.

R// This passage has been removed based on the reviewer’s comments about the model.

L164: Is the sperm motility scale quantifying sperm transfer, sperm viability, or both? Were you able to get a sense if one or the other is especially important in PMPZ?

R// The particular assay the reviewer is referring to only assessed the movement of sperm tails by visual inspection under the microscope. This is more clearly stated now in Lines 165-168. We did not quantify the amount of sperm transferred by males during copulation.

L175: Why were these two time points selected?

R// These time points were selected based on a previous study (Bono et al. 2011). This information has now been clarified in methods (Lines 178-179)

L176-177: It's a little unclear where the sample number comes from; this could be more explicit.

R// Agreed. We have now included a clearer description of the experimental design (Lines 179-182).

L179: I could use a figure referenced around here that illustrates the experimental design and the DE comparisons.

R// Agreed. A new figure showing the experimental design has now been included (new Fig 1).

L219: I'm not clear on the relationship between DE and AS genes. I initially thought AS was a subset of DE, but now I'm not sure. It seems AS genes are identified if splicing happens at a different rate post-copulation. This criterion would not detect genes that are spliced differently in the female reproductive tract relative to other tissues. This needs to be clarified.

R// Correct. Our AS analyses do not identify genes that are spliced differently in the female reproductive tract relative to other tissues. We identified genes that are alternatively spliced in response to mating relative to virgin females based on exon and splice junction usage (relative to overall expression of the gene). We have reinforced this explanation in our methods (Lines 238-241).

L301: delete "postmating".

R// Agreed and changed.

L312: depending on journal conventions, it's helpful to have figure legends that do more than just describe the content of the figure. What is the conclusion to be made?

DE genes were more likely to be conserved no overlap in GO function between species suggested functional divergence.

R// As far as we can gather there is no journal rule indicating the need to include conclusions in the figure legends, nor does it explicitly state that we are not allowed to do so. A cursory review of the most recent publications in CommBio shows that typically a conclusion is not present in the figure legends. Yet, we agree with the reviewer that in some figures this would provide more clarity. Hence, when appropriate, we have changed the figure legends (see Figures 2, 3, 5).

L335: Fig. 2 and elsewhere - avoid red-green color schemes. It's helpful to the reader to have the same meanings for colors used consistently throughout. For example, here, green is both conspecific and AS, and red is both heterospecific and DE. Orange here is DE-AS-IR, and also used in Fig 5 for both DE and AS. Are all these genes DE, but not all are AS or IR? It seems that IR is a subset of AS, which is a subset of DE, so these are not necessarily mutually exclusive categories. If so, make sure this is clear.

R// Thank you for the comment. We have recolored all the figures in the manuscript to make sure they will be discernable to color blind readers. Additionally, as suggested by the reviewer we have made sure that the color scheme is consistent throughout the manuscript.

L352: "was not explained by the regression model" – are these results reported somewhere?

R// These results refer to the R^2 estimated from Pearson's correlation coefficients, which are indicated in Figure 3 (now Figure 4). The R^2 represents the variation explained by the model while we are discussing the complement of R^2 ($1-R^2$), the variation not explained by the model.

L444: Fig 7 – Does All data mean all genes? There don't seem to be enough black dots for that.

R// We have clarified the legend of Fig 7 (now Fig 8) to better explain what we meant by "all data". As stated in the legend now "The black line shows the overall correlation using all mating responsive genes. "

L448-449: I'm confused why this statement is necessary.

R// We believe this statement is necessary to make the reader aware that only significantly differentially expressed genes were used in the shown correlations.

L450-1: this description is different from the legend. Also, separate out (blue) and (red) to be after their respective descriptions.

R// We changed the Fig 7 (now Fig 8) legend to now state, "The blue line corresponds to the subset of genes that was DE in the conspecific cross. The red line corresponds to the subset of genes that was DE in the heterospecific cross but not the conspecific cross."

L463: I'm very confused by Fig 8. These are genes with disrupted expression profiles compared with the genome background (presumably genes that were not DE?) - so which genes are the DE genes and which genes are not? Does con vs het here (red vs green) indicate disrupted genes that are normally DE in conspecific matings vs DE genes that are only DE in het matings? The caption doesn't really describe this. Do the *** indicate significance between green and red? or between Dmoj and Dari? The caption doesn't make it clear that these genes are the ones that are different between con and het matings (if that's what they are).

R//All the genes used in this analysis had disrupted transcriptional profiles in heterospecific matings. The conspecific disrupted genes are genes that responded to conspecific mating but were disrupted in heterospecific matings. The heterospecific disrupted genes were those that were only differentially expressed in heterospecific matings. We have clarified these points by edits to the text and the figure legend. As described in the legend, the ** indicates significance of comparisons against the genome background.

L476: add “in” before “oviposition”.

R// Agreed and changed (Line 498).

L510: “more comprehensive transcriptomic response” – be more specific, what do you mean exactly? It’s actually stated in the second half of the sentence – this should be said at the beginning.

R// Agreed and changed (Lines 528-530).

L521-523: citation?

R// A citation has been included for this claim (Line 541).

L559-561: confusing sentence.

R// Agreed. The sentence was rephrased for clarity in the corresponding section (Lines 575-576).

L572-576: confusing sentence.

R// We have rephrased this section to provide more clarity (Lines 596-603).

L577-583: This is not the conclusion I would have gleaned from the above sentences. It sounds like you’re trying to say that immune genes are misregulated, but then you say that there isn’t strong evidence for heightened immune response after heterospecific crosses. I don’t know why downregulation of immune genes would be inconsistent with the idea that the immune response is misregulated, since misregulation is not direction-specific.

R// We agree that our logic was not clear in this section. The point we are trying to make is that the enrichment analysis makes it appear like immunity is misregulated in heterospecific matings. However, just because a functional cluster is enriched in one cross and not the other does not necessarily mean that overall patterns of expression are different. Whether a category is overrepresented or not depends on the overall set of DE genes identified, which differs for each cross. We have tried to clarify this point the revised manuscript (Lines 592-603).

L585-586: this seems like a reach unless you can further justify this line of thinking with other evidence.

R// We have added some additional evidence that further justifies including this section, especially given the reviewer's suggestion above of including more speculation on the importance of individual genes. We also added language to make it clear that our conclusions are speculative (Lines 610-620).

L609: "transcriptional divergence and heterospecific disruption" – explain again what this means.

R// Agreed. We have clarified the meaning of this concept in the corresponding section (Lines 639-642). We also note that we slightly changed our metrics for transcriptional divergence and heterospecific disruption from the first submission, by now taking the absolute value of calculated differences (as described in Lines 277-283 and the legend for Fig 8). This changes the appearance of the figure, but not the overall conclusions we draw from the analysis.

L609: edit "all" to "both".

R// Agreed and changed.

L624-626: I didn't get this from Fig 8, but I was also confused by this figure. Doesn't this go against the coevolutionary divergence model?

R// We agree, and we hope this statement is clearer with the new changes introduced to the legend of Fig 8 (now Fig 9).

L626-628: was this shown somewhere? I'm assuming categories means DE, AS, IR?

R// This is shown in Figure 9. As described above, we have edited the figure to make it clearer what is being compared.

L635-637: this is a good argument.

R// We thank the reviewer for the positive comment.

L642-643: the second half of this sentence is confusing.

R// Agreed. The sentence was modified for clarity.

L654-657: this last sentence is too long and needs to be streamlined.

R// Agreed. We have modified the long sentence for clarity (Lines 681-685).

REVIEWERS' COMMENTS:

Reviewer #4 (Remarks to the Author):

I thank the authors for addressing my concerns. I think the paper is now suitable for publication.

Response to reviewers

We would like to thank the editor for considering our manuscript for publishing in *Communications Biology*, as well as the previous reviewer and a new reviewer for their valuable suggestions. We have carefully read all suggestions and questions raised by the reviewers, which have helped us to improve the revised version of our manuscript. Here we explain how we responded to these suggestions point by point. In our submitted revised manuscript, we have highlighted in yellow all the sentences that were modified as per the reviewers' suggestions.

Reviewer #2 (Remarks to the Author):

The authors addressed all my concerns in this revised version of the manuscript.

We are happy to hear that all the suggested improvements to our manuscript were acceptable. Thank you for your comments.

Reviewer #4 (Remarks to the Author):

First, I found the paper difficult to read and (like Reviewer 3) had to reread many sentences and whole paragraphs to understand what the authors were trying to say. I do not think the authors took this comment as a major concern. They say that “have gone through the entire text and when appropriate have simplified the language” (underlining is my emphasis). I do not know what the first version looked like, but the revision is still hard to follow. I suspect that they did not find it “appropriate” to simplify the language in many places.

We appreciate the careful review and suggestions. We have made an honest effort to make the paper more readable and easier to follow. Specifically, in the methods/results/discussion sections, we have added statements that clarify the purpose of each analysis, and we summarize the main conclusions at the end of paragraphs. We tried to be more consistent in our terminology and language throughout, particularly with reference to the different gene sets (see below). We believe a lot of the confusion stems from the way we originally presented some of the data, which is addressed below.

Also regarding “major concern 1”, I agree with Reviewer 3 that the results are hard to follow. The authors say they added a new figure 1 outlining the experimental design, but this figure does not help to reduce the cognitive load. I do not think this figure is necessary because this aspect of the experimental design is easy to follow.

We opted to keep this figure since it was a clear suggestion by a previous reviewer. We agree with this suggestion and feel that some readers may benefit from the visual representation of the experimental design. We also added information in the legend of Figure 1 describing the comparisons. If the reviewer feels strongly that it should be removed, we will do so.

The complexity that is hard to follow (and that I imagine Reviewer 3 is referring to) is keeping track of which subsets of DE (or AS) genes are being analyzed. For example, in Figure 3, what are the conspecific only, heterospecific only, and con- and heterospecific columns referring to? Do these refer to the different sets from the Venn diagram on the right? Is it appropriate to lump them together in a single column if the DE and AS lists come from different datasets? The figure legend does not help. Panel A does not simply report the number of DE, AS, and IR genes following con- and heterospecific matings. (That figure would just have two columns: one for the conspecific matings and one for the heterospecific matings.) These data have been subsetted further, but it's hard to follow exactly how.

We thank the reviewer for raising this concern. We agree that more detailed explanations and figures will help the reader to understand the complexity of this section in the results. To address the reviewer's concern, we have made substantial changes to the way the analyses are described in the methods, and the data are presented in the results. Specifically, we have made the following major changes:

Methods:

We made extensive changes to the section "*Postmating transcriptional divergence and disruption of gene expression in heterospecific matings.*" We added sections that describe in more detail how we identified the different gene sets used in the analyses. We also give more detailed explanations for the purpose of each scatterplot and which gene sets were used in each.

Figures:

We believe figure 3a was the source of some of the confusion because the original version was a summary of all the transcriptomic data, but, as pointed out by the reviewer, was not presented in a way that made it apparent where different sets of genes used in subsequent analyses came from (e.g. "conspecific mating responsive" genes or "heterospecific only" genes). In the revised version, we have split up this information across multiple figures and presented the data in a way that should be much easier to ascertain where the different gene sets come from. Specifically, the new figure 3 focuses only on the conspecific crosses. Panel 3a presents the data on the conspecific mating response in each species (as suggested by the reviewer), while panels b, c and d compare the conspecific crosses between the species. With regard to the reviewer's concern about stacked bars, we note that the DE and AS lists come from different analyses of the same datasets. Each differentially regulated gene identified in these analyses falls into one of the four categories shown, and together they represent the total number of differentially regulated genes. The stacked bar format makes it easy to compare the relative contribution of the different forms of gene regulation to the overall postmating response in each species. We therefore believe it is appropriate to present the data in this format.

We made two new figures that present data on the heterospecific crosses (figure 6 and supplementary figure S4). The Venn diagrams presented in figure 6 make it easy to identify the

different gene sets used in subsequent analyses (e.g. “conspecific mating responsive”, “heterospecific only” and “misregulated mating responsive”).

We added supplementary table 1, which summarizes the number of genes that were differentially regulated in all cross/time combinations, and a supplementary file that presents all the analyses outputs for DE, AS and IR.

We removed the original figure 6 and associated analyses in an effort to simplify the paper. These analyses were challenging to describe and interpret, and the main conclusions can be drawn from other figures.

We edited all figure legends for clarity. We specifically identify which genes belong to the different gene sets we discuss (e.g. “conspecific mating responsive, “misregulated mating responsive”, “heterospecific” only, etc.).

Similarly, I found all of the scatterplots hard to follow. How are Figures 4, 7, and 8 substantially different from each other and how were the blue and yellow subsets identified? As recommended by the reviewer, it would be helpful to have some figure explaining how the conspecific mating responses versus heterospecific only genes were identified.

As mentioned above, we have provided more information on the scatterplots in the methods (lines 270-302). We also added figure 6, which explains how the blue and yellow gene sets were identified. We also added information to the legends that further clarify the color scheme.

For conspecific expression divergence and heterospecific disruption, is it appropriate to calculate these using absolute values? Seems like you’re losing potentially important information by forcing all of these comparisons to be positive.

We used the absolute value because our hypothesis was that the magnitude of transcriptional divergence would be positively correlated with the magnitude of transcriptional disruption irrespective of the direction of transcriptional changes. As an example, if the conspecific expression change in *D. arizonae* is higher than *D. mojavensis* and the difference is large, we predict that disruption in the heterospecific crosses would be high as well. The hypothesis does not predict whether the expression change in the heterospecific cross should be higher or lower than that in the conspecific cross, just that the magnitude of the difference should be positively correlated with the magnitude of divergence in the conspecific crosses. We have added this justification to the text (lines 312-317).

Regarding “major concern 2”, I think there may still be remnants of the old model. On line 62, I do not know what “this model” is referring to. Perhaps the coevolutionary divergence model that has now been (mostly) deleted? I agree with Reviewer 3 that the co-evolutionary divergence model is not unique and doesn’t need a special name.

We changed the wording as requested.

Line 179: the authors did not address the concern about DE comparisons. This aspect of the study is still very difficult to follow (see comment above).

We addressed this with the changes we detail above to the methods and results.